# A Comparison of Ethylene-Tar-Derived Isotropic Pitches Prepared by Air Blowing and Nitrogen Distillation Methods and Their Carbon Fibers

**DOI:** 10.3390/ma12020305

**Published:** 2019-01-18

**Authors:** Kui Shi, Jianxiao Yang, Chong Ye, Hongbo Liu, Xuanke Li

**Affiliations:** 1College of Materials Science and Engineering, Hunan University, Changsha 410082, Hunan, China; skhnu123@163.com (K.S.); 15874950624@163.com (C.Y.); hndxlhb@163.com (H.L.); 2Hunan Province Key Laboratory for Advanced Carbon Materials and Applied Technology, Hunan University, Changsha 410082, China

**Keywords:** carbon fiber, ethylene tar, isotropic pitch, air blowing

## Abstract

Two isotropic pitches were prepared by air blowing and nitrogen distillation methods using ethylene tar (ET) as a raw material. The corresponding carbon fibers were obtained through conventional melt spinning, stabilization, and carbonization. The structures and properties of the resultant pitches and fibers were characterized, and their differences were examined. The results showed that the introduction of oxygen by the air blowing method could quickly increase the yield and the softening point of the pitch. Moreover, the air-blown pitch (ABP) was composed of aromatic molecules with linear methylene chains, while the nitrogen-distilled pitch (NDP) mainly contained polycondensed aromatic rings. This is because the oxygen-containing functional groups in the ABP could impede ordered stack of pitch molecules and led to a methylene bridge structure instead of an aromatic condensed structure as in the NDP. Meanwhile, the spinnability of the ABP did not decrease even though it contained 2.31 wt % oxygen. In contrast, the ABP had narrower molecular weight distribution, which contributed to better stabilization properties and higher tensile strength of the carbon fiber. The tensile strength of carbon fibers from the ABP reached 860 MPa with fiber diameter of about 10 μm, which was higher than the tensile strength of 640 MPa for the NDP-derived carbon fibers.

## 1. Introduction

Carbon fibers (CFs) are widely used in the military, various industries, and sports because of their high mechanical properties, low density, and good conductive properties [1]. The raw materials of pitch-based CFs are usually coal or petroleum-derived by-products, which are abundant, low cost, and have high carbonization yield [2,3]. Therefore, more and more researchers are pursuing low-cost, pitch-based CFs because of their considerable advantage in price and extensive application prospects in the fields of automobiles, sporting goods, building materials, C/C composites, activated carbon fibers, thermal field, etc. [4,5,6,7,8,9]. Based on this, a lot of raw materials and preparation methods of pitch precursor have been tried, including bromination and subsequent dehydrobromination of naphtha-cracked oil [3], tailored suitable molecular weight portion from Hyper-coal by methylnaphthalene [10], heat treatment of pyrolyzed fuel oil [11], and so on. Ethylene tar (ET) is the by-product of ethylene production, which is rich in resources, low cost, and has low ash content compared to other precursors, especially coal tar pitch, which usually contains primary quinoline insoluble that needs to be removed first. Accordingly, ET is expected to be an ideal raw material for preparing a spinnable pitch. In addition, atmospheric distillation and air blowing are the most common methods to prepare a spinnable pitch for CF production [12,13,14,15,16]. Compared to atmospheric distillation, which mainly involves the removal of light components and condensation polymerization of heavy components during the reaction, air blowing is recognized as an effective method to increase the softening point (SP) and the yield of the pitch. This is because oxidation can link molecules by methylene formed through oxidative dehydrogenation of aliphatic side chains or by oxygen-containing functional groups, such as C–O–C and C=O bridge, formed through oxidation of aliphatic side chains [12,15,17]. Consequently, air blowing can also suppress the formation of mesophase to obtain a homogenous pitch [18]. There has been some research on air blowing of pitches, particularly focusing on the influence of different raw materials, such as coal tar pitch, petroleum pitch, and anthracene oil, or different conditions of air blowing on the final properties of air-blown pitches (ABP) [19,20,21]. However, these works only investigated the pitches and rarely referred to the preparation of CFs. As a matter of fact, when preparing CFs, oxygen is introduced not only during the air blowing process but also during the stabilization process in order to make the pitch fibers infusible and maintain the fiber shape during the carbonization process. The oxygen introduced in these two stages may have different behaviors during subsequent carbonization processes. Indeed, it is important to consider the oxygen that exists in the pitch precursor as it may lead to a different molecular structure, and the corresponding molecular structure of the pitch would then impact the spinnability of pitches as well as the stabilization and carbonization processes of pitch fibers. In this work, we propose that the oxygen introduced in the pitch precursor can improve the mechanical properties of the resultant CFs prepared at low carbonization temperature below 1200 °C because the introduced oxygen in the pitch precursor is more stable than the introduced oxygen from the stabilization process.

Two kinds of ET-derived isotropic pitches were prepared by nitrogen distillation and air blowing methods in order to compare the differences in their molecular structures and clarify the influence of oxygen introduced during the preparation of the pitch precursor or during the stabilization process of pitch fibers on the properties of the corresponding CFs.

## 2. Materials and Methods 

### 2.1. Materials

ET was supplied from Wuhan Luhua Yueda Chemical Co. Ltd (Wuhan, China). The ET was used as a raw material to prepare the spinnable pitches; ET is completely soluble in toluene.

### 2.2. Preparation of Spinnable Pitches

The air-blown pitch was prepared by the air blowing method as follows: (1) The ET was distilled at 250 °C in a 2 L stainless steel reactor to remove the light components to obtain a basic pitch. (2) The basic pitch was air blown at 280 °C for 3 h in 3 L/min air atmosphere to attain oxidized pitch. (3) In order to get the spinnable pitch with high softening point (SP), the oxidized pitch was further heat-treated at 350 °C for 4 h in 3 L/min nitrogen atmosphere. For comparison, the nitrogen-distilled pitch (NDP) with almost the same SP was prepared by heating the ET at 380 °C for 5h in 3L/min nitrogen atmosphere.

### 2.3. Preparation of Carbon Fibers

The prepared NDP and ABP were spun into pitch fibers (PFs) at a temperature equal to their SP +80 °C using a melt-spinning method with a single-hole spinneret (diameter = 0.2 mm, length/diameter = 3). For this procedure, a laboratory spinning apparatus (Huizhong Dingcheng, Chengdu, China) was used with a nitrogen pressure of 0.4 MPa and winding speeds of 300–500 rpm (100 rpm = 60 m/min) to get PFs with different diameters. The spun PFs were stabilized by heating from room temperature to 280 °C at a rate of 0.5 °C/min and then holding at this temperature for 1 h with an air flow of 500 mL/min. Then, the stabilized fibers (SFs) were successively carbonized at 1200 °C for 30 min at a heating rate of 5 °C/min with a nitrogen flow rate of 100 mL/min. The resultant PFs, SFs, and CFs from the NDP and the ABP were labeled as NDP-PF, ABP-PF, NDP-SF, ABP-SF, NDP-CF, and ABP-CF, respectively.

### 2.4. Characterization of Pitches and Carbon Fibers

The SP of the pitch was determined by a CFT-100EX capillary rheometer (Shimadzu, Kyoto, Japan). The solubility of the pitch in *n*-hexane, toluene, and quinoline was determined using the Soxhlet extraction method (GB/T 26930.5-2011) to obtain *n*-hexane soluble (HS), *n*-hexane insoluble and toluene soluble (HI-TS), toluene insoluble and quinoline soluble (TI-QS), and quinoline insoluble (QI) fractions. Carbon, hydrogen, sulfur, and nitrogen contents were determined according to SN/T 4764-2017 with a Elementar Vario EL III elemental analyzer (Elementar, Langenselbold, Germany). The oxygen content was obtained by the subtraction method (O=100−C−H−N−S). Fourier transform infrared (FT-IR) spectra were obtained using the KBr disc technique (sample/KBr = 1/100) in a Nicolet iS10 FT-IR spectrometer (Thermo Fisher Scientific, Waltham, MA, USA). Each spectrum was an average of 32 scans with a resolution of 4 cm^−1^. The solution-state ^13^C nuclear magnetic resonance (^13^C-NMR) spectra were obtained using a Bruker 600 MHz Advance NMR spectrometer (Bruker, Karlsruhe, Germany). The quantitative ^13^C-NMR spectra were recorded by dissolving samples in *d*-chloroform (CDCl_3_) solvent (sample/CDCl_3_ = 100 mg/1 mL) with tetramethylsilane used as the chemical shift reference. The ^13^C-NMR spectra were quantitatively analyzed by the ratio of the peak integral area. X-ray diffraction (XRD) analyses were performed by a D8 Advance diffractometer (Bruker, Karlsruhe, Germany) with Cu Kα radiation (λ = 0.15406 nm) generated at 32 kV and 50 mA with a scan speed of 1 °/min for 2 theta values between 5° and 80°. The thermogravimetric (TG) properties of samples were measured using a STA 449 F5 thermal analyzer (Netzsch, Selb, Germany). The thermal stability and coking value (CV) of the obtained pitches were analyzed in 40 mL/min nitrogen atmosphere with a heating rate of 5 °C/min to 900 °C. Meanwhile, to evaluate the stabilization properties of spun PFs, the obtained PFs were also analyzed by TG with different heating rates (0.5 °C/min, 1 °C/min, 2 °C/min, 4 °C/min) to 600 °C in 40 mL/min air atmosphere to find their maximum weight gain (W_max_) and the corresponding maximum temperature (T_max_) as well as to calculate their reaction activation energy (E_a_). The gas released during the carbonization of PFs and SFs were measured by Hiden Analytical HAS-301-1474 mass spectrometer (Hiden Analytical, Warrington, UK) coupled with TG, which was heated from 20 to 1200 °C at a rate of 10 °C /min and an argon flow of 20 mL/min. The MS was performed at RGA mode with a secondary electron multiplier, and the quartz capillary connected to the thermal analyzer was heated to 160 °C. The morphologies and the diameter of the CFs were observed by JSM-6700F field emission scanning electron microscope (SEM, JEOL, Tokyo, Japan) with 5 kV. The tensile strength and Young’s modulus of CFs were measured at room temperature using monofilaments with a gauge length of 20 mm according to the standard (ASTM D4018-2011). The diameter of CFs was observed by SEM after the tensile experiment. The tensile strength and Young’s modulus were evaluated from the mean value of 30 tests, with the values distributing within 10%.

## 3. Results and Discussion

### 3.1. Characterization of NDP and ABP

The general characteristics of pitches are summarized in Table 1. The yield of the ABP was higher than that of the NDP, and their SP was almost the same. This result is in line with results from previous researches because the air blowing method can increase SP more quickly [17]. It was apparent that air blowing brought large amounts of oxygen into the pitch as the oxygen content of the ABP was up to 2.31%. This was higher than the oxygen content of the NDP, which was 0.78%. Compared to the ET, the solubility of the NDP and the ABP in *n*-hexane and toluene decreased; HI-TS increased from 48.4% for the ET to 59.8% and 83.2% for the NDP and the ABP, respectively, as shown in Figure 1. It should be noted that TI appeared in both the NDP and the ABP, but the TI of the NDP was larger than that of the ABP. In addition, QI appeared in the NDP, which demonstrated that the nitrogen distillation method was more effective in accelerating the polymerization reaction and forming larger molecules than the air blowing method [22]. This could also be confirmed by the larger C/H atom rate of the NDP compared to the ABP, i.e., 1.40 and 1.26, respectively. Although the NDP had larger molecules, it is believed that the molecular weight distribution of the ABP was narrower and more uniform than the NDP due to its quite high HI-TS content. The SP of ABP was equal to that of the NDP, even at lower treatment temperature, because larger molecules were produced through oxidative cross-linking of small molecules when air blowing. Meanwhile, oxidative cross-linking suppressed molecule aggregation and uneven polymerization that might happen in the nitrogen distillation process. This more homogeneous composition of ABP would be beneficial for its spinning performance. 

FT-IR analyses, shown in Figure 2, were carried out to verify the functional groups in the NDP and the ABP. The absorption peaks at 3050 cm^−1^ and 1600 cm^−1^ were assigned to the presence of aromatic C–H and aromatic C–C stretching vibrations, respectively. The stronger absorbance at 2920 cm^−1^ and 2850 cm^−1^ in the ABP corresponded to methylene hydrogen asymmetric and symmetric stretching vibrations, respectively [23]. The peak at 1450 cm^−1^ corresponding to methylene hydrogen bending vibration was clearly strong in intensity. The broad peak at 3300–3600 cm^−1^ might be ascribed to hydroxyl of H_2_O in the pitches [24]. It must also be mentioned that a new peak appeared at 1700 cm^−1^ for the ABP compared to the NDP, which was assigned to the C=O stretching vibration. This could be easily explained by the higher oxygen content of the ABP, as shown in Table 1. Another weak bond at 1260 cm^−1^ belonging to the C–O stretching vibration could also be observed. The emergence of oxygen-containing functional groups indicated that oxygen might connect pitch molecules as an oxygen bridge by air blowing.

In order to further study the molecular structure of the NDP and the ABP, ^13^C-NMR was performed. The ^13^C-NMR spectra of the NDP and the ABP are plotted in Figure 3. The normalized integration data of ^13^C-NMR spectra is presented in Table 2. The larger aromaticity of the NDP could be demonstrated by larger C_ar_/C_al_ compared to the ABP, i.e., 5.29 and 2.85, respectively. Most of the aromatic carbon in NDP was C_ar1,3_, suggesting abundant pericondensed structure in the NDP [25]. This result indicated more condensed large molecules in the NDP produced by heat treatment, which was reflected by the higher insoluble content, as shown in Figure 1. More aliphatic carbon in the form of CH_2_ and C_α2_ in the ABP compared to the NDP indicated more methylene bridge structures in the ABP, formed by oxidative dehydrogenation, which is in accordance with the above analysis.

Figure 4 shows the XRD graphs of the NDP and the ABP. Both of them showed broad peaks between 10° and 30°. Peaks between 10° and 20° were attributable to asphaltene components, while those between 20° and 30° were attributable to stacked molecular structures [26]. Therefore, the stacked structure was more evident in the NDP than that in the ABP due to larger condensed aromatic molecules in the NDP, as shown in Figure 1. In contrast, the oxygen in the ABP impeded ordered stack of molecules. Nevertheless, more symmetrical peak of ABP may indicate more uniform composition, which is consistent with the extraordinarily uniform components presented in Figure 1.

### 3.2. Spinning Properties of NDP and ABP

The viscosity–temperature curves of NDP and ABP are shown in Figure 5. The viscosity of both the NDP and the ABP decreased sharply with increasing temperature when the temperature was lower than 310 °C, then decreased gradually to about 330 °C, while the viscosity–temperature curves became almost flat above 330 °C because of the temperature sensitive property of the pitch. It can be seen that the viscosity–temperature curve of the ABP showed two jumping steps between 310 °C and 330 °C, which suggested that the ABP had worse spinning performance than the NDP. However, both of them had excellent spinning performance when the NDP and the ABP were spun into PFs by the melt-spinning method at 335 °C. Therefore, the results indicated that the spinnability of the ABP had not deteriorated even though it had more oxygen, which is usually considered as an impurity atom. This may be attributed to the homogenous components of the ABP, as previously mentioned. 

### 3.3. Stabilization and Carbonization of NDP-PF and ABP-PF

Stabilization is a crucial process to determine the properties of CFs due to the formation of intermolecular cross-linking in an oxidizing atmosphere, which can ensure that the shape of fibers do not change in subsequent carbonization processes [27]. Therefore, proper stabilization parameter should be adopted. In order to optimize the stabilization process, TG was used to measure the oxidation reactivity of PFs, and the results are presented in Figure 6. The lower initial temperature of weight gain indicated higher oxidation reactivity of ABP-PF compared to NDP-PF. This was also proven by the lower E_a_ and T_max_ of ABP-PF, as shown in Table 3. The higher oxidation reactivity of ABP-PF might have been due to the lightest components remaining in the ABP instead of being removed by distillation at high temperature as in the NDP. However, the W_max_ in oxidation of ABP-PF was less than NDP-PF, i.e., 11.6% and 12.8%, respectively, as exhibited in Table 3. This result could be interpreted as showing that the higher oxygen in the ABP could restrain more oxygen from diffusing into the PFs. This viewpoint can be demonstrated by the similar oxygen content of NDP-SF and ABP-SF, as presented in Table 3.

After 1200 °C carbonization, both NDP-CF and ABP-CF had high yield over 70%, which is favorable for producing low-cost, general-purpose CFs. Higher oxygen content of ABP-CF compared to NDP-CF, as shown in Table 3, denoted that less oxygen of ABP-CF was given off during the carbonization process in view of the similar oxygen content of NDP-SF and ABP-SF. Less gas was released in the carbonization process of ABP-SF (35.7%) compared to the NDP-SF (37.7%), which might be beneficial for the mechanical properties of CFs because fewer defects would be generated. This also illustrates that the oxygen existing in the pitch precursor was more stable than that introduced in the stabilization process. This could be verified by the TG–MS results of the CO and CO_2_ given off from oxygen-containing groups of fibers during carbonization process for NDP-SF and ABP-PF, as shown in Figure 7a,b. Both CO and CO_2_ were released at two stages; the first peak was located at 400–800 °C and 270–850 °C, respectively. Then, the amount of release began to increase rapidly for both CO and CO_2_, especially for NDP-SF. It should be noted that the difference in the release amount of CO_2_ for NDP-SF and ABP-PF was more distinct. The NDP-SF revealed higher peak intensity and magnitude of CO_2_ in both stages, especially the high temperature stage, compared to ABP-PF. This showed that the removal of oxygen was more arduous below 1200 °C when the oxygen was from the pitch precursor introduced by air blowing than the oxygen from the stabilization process. Therefore, moderate amounts of oxygen in the pitch precursor could improve the stabilization and carbonization properties of PFs.

### 3.4. Morphology and Mechanical Properties of NDP-CF and ABP-CF

The SEM micrographs of CFs are shown in Figure 8. ABP-CF expressed more homogeneous and smooth surfaces and cross sections compared to NDP-CF, corresponding to more homogeneous precursor pitch. Moreover, the CFs exhibited no fusing, indicating that the stabilization process had been adequate. The cross section of the CFs showed a glass-like fracture surface, indicating that the precursor pitch was isotropic. The tensile strength and Young’s modulus of CFs are presented in Figure 9. ABP-CF had a tensile strength of 550–895 MPa and modulus of 31–45 GPa when the diameter decreased from 14.1 to 9.8 μm; these figures were consistently higher than for NDP-CF, which had tensile strength of 448–748 MPa and modulus of 27–35 GPa as the diameter decreased from 13.1 to 8.7 μm. In both cases, the tensile strength decreased with increasing diameter. The mechanical properties reached the level of commercial, general-purpose CFs. Higher tensile strength can be attributed to the uniform composition of the ABP and less gas released in the carbonization process, which can produce defects in CFs due to the existence of oxygen, as previously discussed. Whether the greater amount of oxygen, which would be removed at higher temperature above 1200 °C had an unfavorable effect on the properties of CFs needs further investigation.

## 4. Conclusions

Two isotropic pitches were prepared through atmospheric distillation and air blowing, respectively. The ABP had a higher yield and more homogeneous composition than the NDP because oxygen could connect smaller molecules and suppress the ordered stack of larger molecules. The oxygen existing in the pitch precursor was harder to remove during carbonization, contributing to higher tensile strength of ABP-CF compared to NDP-CF. The tensile strengths of ABF-CF and NDP-CF were 860 MPa and 640 MPa, respectively, when the diameter was about 10 μm, and the modulus was 41 GPa and 33 GPa, respectively. The results revealed that the oxygen introduced during the pitch precursor preparation process made a significant difference compared to the PF stabilization process. The evolution of introduced oxygen during the pitch precursor, stabilization, and carbonization processes will be investigated in detail in future work.

## Figures and Tables

**Figure 1 materials-12-00305-f001:**
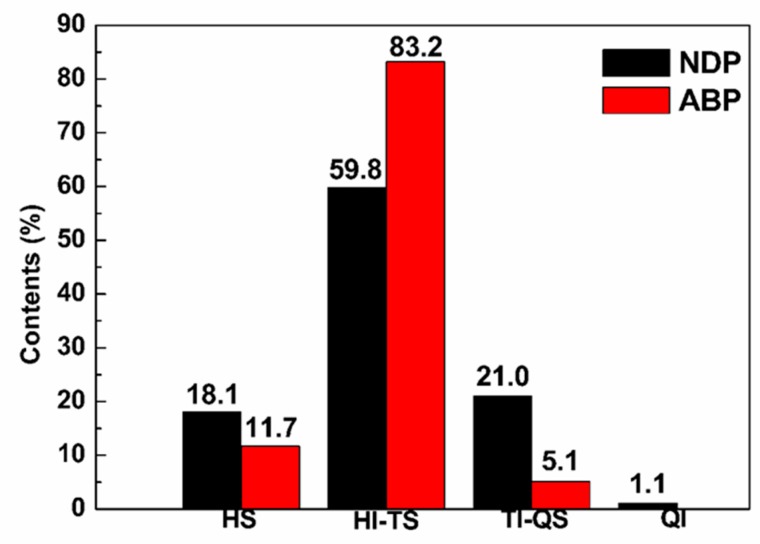
Solubility parameter (*n*-hexane soluble (HS), *n*-hexane insoluble and toluene soluble (HI-TS), toluene insoluble and quinoline soluble (TI-QS), and quinoline insoluble (QI)) of the NDP and the ABP.

**Figure 2 materials-12-00305-f002:**
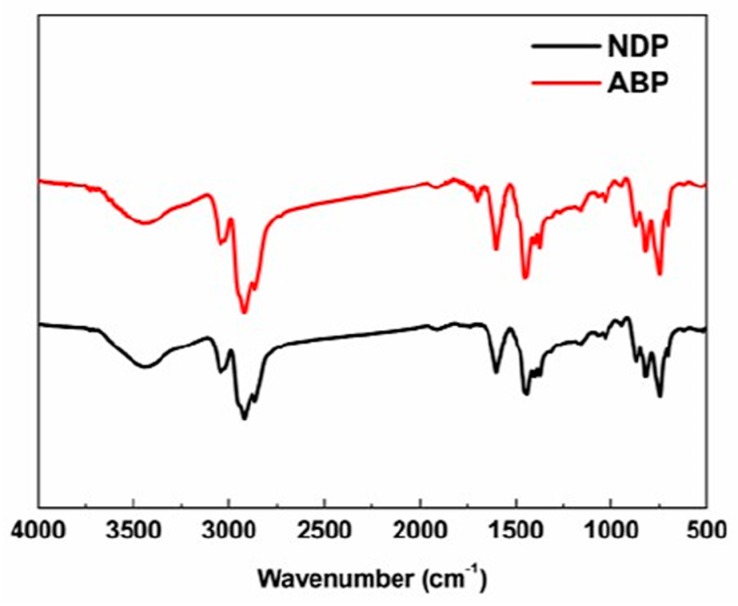
FT-IR spectra of the NDP and the ABP.

**Figure 3 materials-12-00305-f003:**
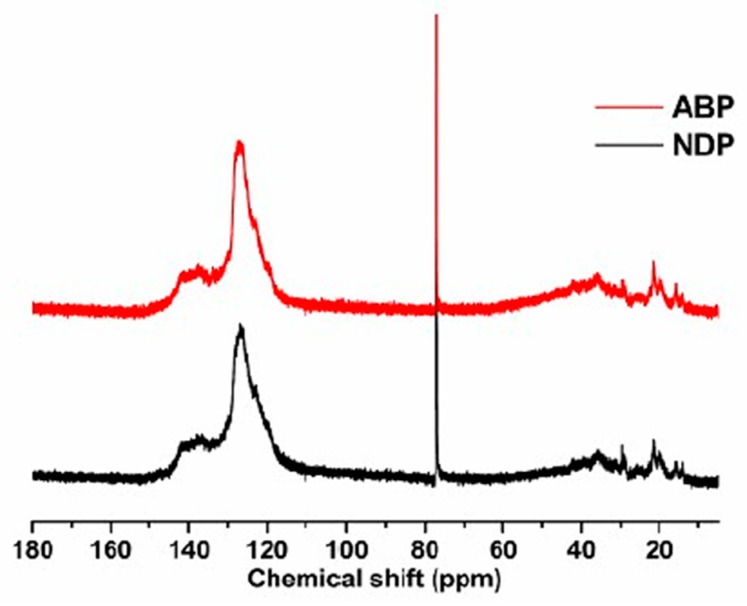
^13^C-NMR spectra of the NDP and the ABP.

**Figure 4 materials-12-00305-f004:**
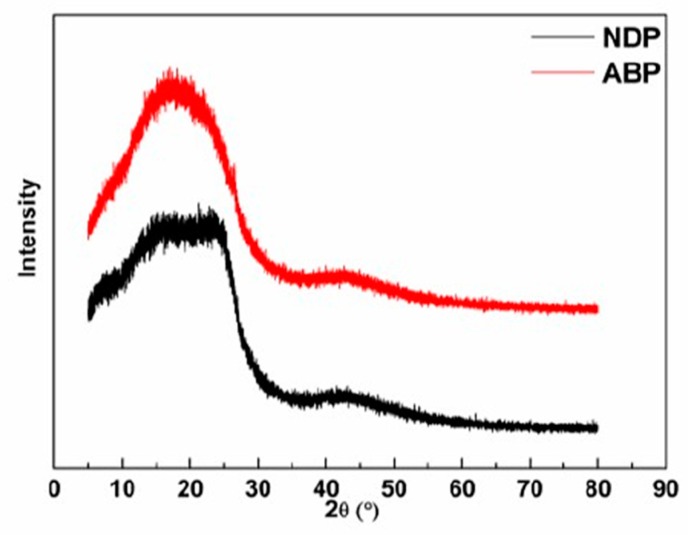
X-ray diffractograms of the NDP and the ABP.

**Figure 5 materials-12-00305-f005:**
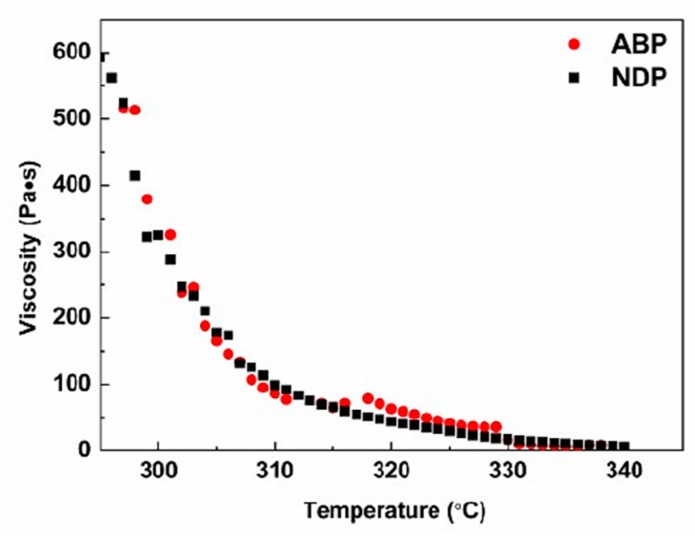
Viscosity–temperature curves of the NDP and the ABP.

**Figure 6 materials-12-00305-f006:**
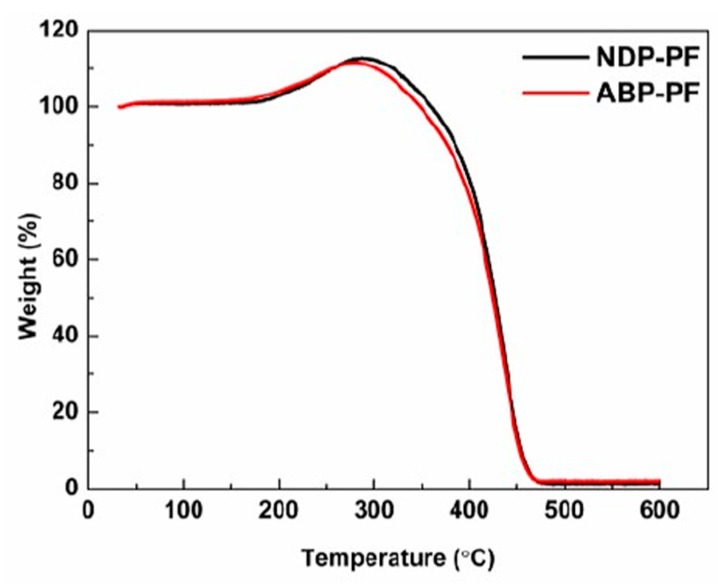
Thermogravimetric (TG) curves of NDP-PF and ABP-PF during stabilization process in air atmosphere.

**Figure 7 materials-12-00305-f007:**
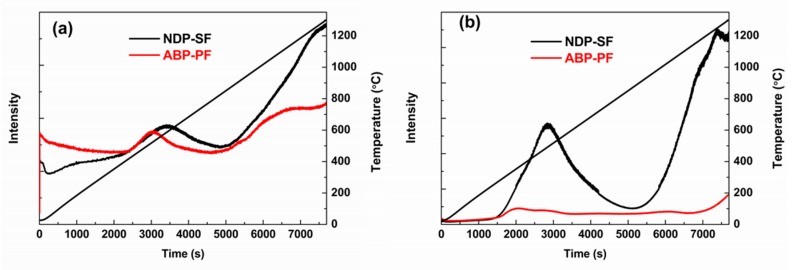
MS curves of NDP-SF and ABP-PF during carbonization process in argon (**a**) CO and (**b**) CO_2_.

**Figure 8 materials-12-00305-f008:**
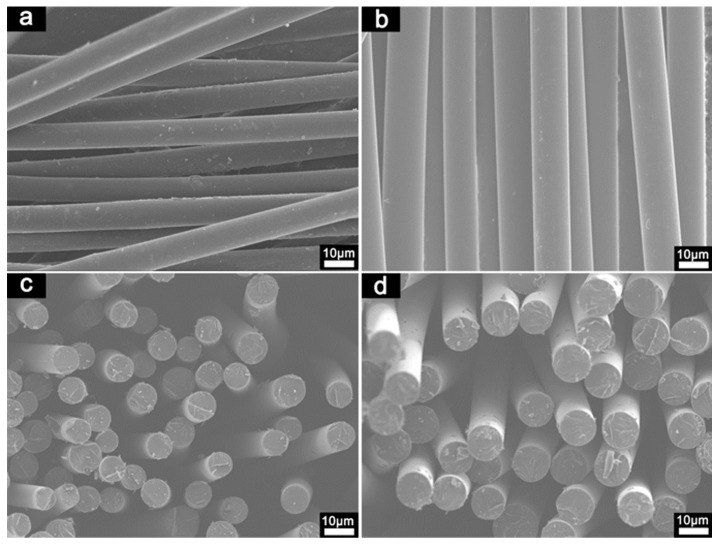
SEM micrographs of (**a**,**c**) NDP-CF and (**b**,**d**) ABP-CF.

**Figure 9 materials-12-00305-f009:**
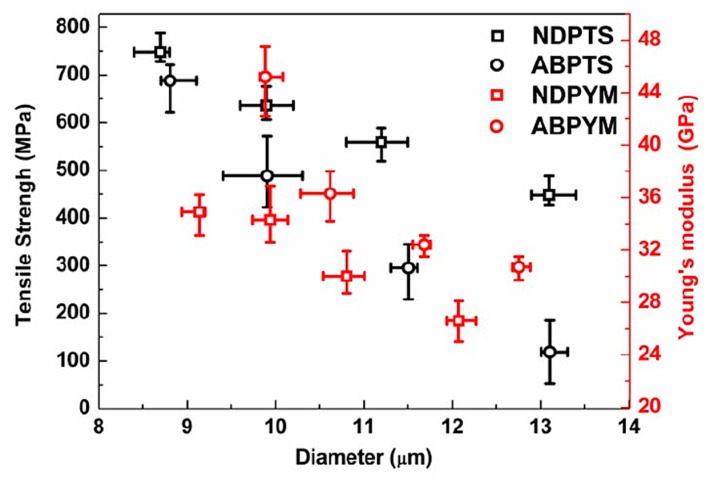
The mechanical properties of NDP-CF and ABP-CF.

**Table 1 materials-12-00305-t001:** Softening point (SP), pitch yield, and elemental analysis results of the air-blown pitch (ABP) and the nitrogen-distilled pitch (NDP).

Sample	SP (°C)	Yield (%)	Elemental Analysis (%)					
			C	H	N	S	O	C/H
NDP	253	22	93.53	5.56	0.02	0.04	0.78	1.40
ABP	252	28	91.55	6.06	0.05	0.10	2.31	1.26

**Table 2 materials-12-00305-t002:** Normalized integration data based on the ^13^C NMR spectra.

Samples	C_al_			C_ar_		C_al_	C_ar_	C_ar_/C_al_
	CH_3_	CH_2_	C_α2_	C_ar1,3_	C_ar1,2_			
NDP	0.04	0.08	0.04	0.60	0.24	0.16	0.84	5.29
ABP	0.03	0.14	0.09	0.52	0.22	0.26	0.74	2.85

CH_3_, methyl carbon; CH_2_, methylene (CH_2_) carbon α or further from an aromatic ring in free side chain; C_α2_, CH_2_ carbon in bridge/hydroaromatic structures; Car_1,3_, pericondensed aromatic carbon (C_ar3_) and protonated aromatic carbon (CH_ar_); Car_1,2_, catacondensed aromatic carbon, aromatic carbon with both heteroatomic or aromatic substituents (C_ar2_), and the region correspondent to aromatic carbon joined to aliphatic chains; C_al_, total aliphatic carbon; C_ar_, total aromatic carbon.

**Table 3 materials-12-00305-t003:** Oxidation and carbonization properties of pitch fibers.

Samples	T_max_ (°C)	W_max_ (%)	E_a_ (kJ/mol)	Oxygen Content (%)	Yield (%)
NDP-SF	286	12.8	124.5	25.46	114.0
ABP-SF	278	11.6	120.5	25.72	112.5
NDP-CF	-	-	-	8.96	71.3
ABP-CF	-	-	-	10.29	71.0

T_max_, W_max_, and E_a_ were calculated from TG results of NDP-PF and ABP-PF in air atmosphere.

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
