# Peer review of "A Comparison of Ethylene-Tar-Derived Isotropic Pitches Prepared by Air Blowing and Nitrogen Distillation Methods and Their Carbon Fibers"

_materials, 2019, doi:10.3390/ma12020305_

Reviewer 1 Report

See attached file.

Author Response

Response to reviewer 1

Dear Editor

• Manuscript No.: Materials-413012

• Manuscript Title: A comparison of ethylene tar-derived isotropic pitches prepared by air blowing and nitrogen distillation method and their carbon fibers

• Manuscript Authors: Kui Shi, Jianxiao Yang*, Chong Ye, Hongbo Liu and Xuanke Li*

We would like to express our sincere thanks to the reviewers for their valuable comments and constructive suggestions. We have tried our best to revise our manuscript according to their suggestions. We uploaded the revised manuscript which we would like to be accepted to publish in Materials. Revised parts in the manuscript were highlighted in YELLOW.

Response to Reviewer 1:

We would like to express our sincere thanks to this reviewer for her/his valuable comments and constructive suggestions.

(Comment 1)

Several sentences start with And. This word should only be used to connect two sentences.

(Answer to comment 1)

Thank you very much for kind comment.

We have carefully checked the paper and corrected.

(Comment 2)

Line 14: is it correspondent or corresponding?

(Answer to comment 2)

Corresponding is more appropriate. We have corrected it in the revised manuscript (line 14).

(Comment 3)

Abstract should be in present tense.

(Answer to comment 3)

We have read some papers published on this journal (Materials) recently, many abstracts are in past tense. Thus, we think the past tense is also acceptable as the work has been done.

(Comment 4)

The sentence (lines 18-22) is very difficult to understand for non-native English speakers. Generally, the language needs extensive corrections.

(Answer to comment 4)

Thanks for your comment. We have modified this sentence and made it as easy to be understood as possible (lines 18-22).

(Comment 5)

No error bars are provided in the tables (for example 5.29 ± ?, n = ?). Are the experiments are only performed on one sample of each type?

(Answer to comment 5)

Some experiments and measurements are performed once, but the results are still reliable and representative based on our previous lots of experiments which have good repeatability. As for some data like mechanical properties in Fig 9, 20 valid tests were conducted and error bars were added to ensure the accuracy.

(Comment 6)

Line 71/74: is it “mood” or “mode”?

(Answer to comment 6)

We check the word and consider that it is “mood” which means atmosphere, so it can be used here (line 77/79).

(Comment 7)

Obtaining oxygen content by subtraction method may not be accurate in the case of other impurities in the fibers. EDX or other types of analyses must be performed.

(Answer to comment 7)

Thank you for suggested comment. We think it is also reasonable to measure oxygen content by subtraction method through elemental analyses because of the high purity of our raw materials with ash content less than 100 ppm and high precision of elemental analyses. Moreover, many researchers measured oxygen content by subtraction method, for example Ref [10, 20]. As to EDX or XPS, they are also a method to measure oxygen content of materials, but their analysis area is only on the surface of materials with the micron depth. So, we perform to use the elemental analyses method to investigate the oxygen content of pitches and fibers.

(Comment 8)

What is “air moon”? Line 107.

(Answer to comment 8)

Sorry for misspells. It is “air mood” and we have corrected it (line 113).

(Comment 9)

No clear peak is visible in XRD spectra (especially of BP). The broad hump can contain a large variety of carbon materials. The analysis is not done to detail.

(Answer to comment 9)

Thank you for valuable comment. There are just broad peaks in XRD spectra for both NDP and ABP, this is typical amorphous feature of the pitches, but it is obvious that the peak shape is different, which represent different molecular structure in pitches, for example, more condensed structure in NDP due to the stack of lamellar aromatic molecules. However, pitch is a complex mixture of various polycyclic aromatic hydrocarbon and difficult to analyze each molecule in detail. Therefore, we can only obtain synthetical information by XRD, the similar XRD peaks and analysis can be found in Ref [26] (lines 177-183).

(Comment 10)

With the given datapoints in Fig 9 (unlabeled plot of  mechanical strength), no conclusions can be drawn.

(Answer to comment 10)

Fig 9 is the mechanical properties of two carbon fibers. Both tensile strength and Young’ modulus of NDP-CF and ABP-CF increased with the decrease of diameter. ABP-CF has a tensile strength (black circle) of 550 to 895 MPa and modulus (red circle) of 31 to 45 GPa when the diameter decreased from 14.1 to 9.8 μm, always higher than NDP-CF, which was 448 to 748 MPa for tensile strength (black square) and 27 to 35 GPa for modulus (red square) as the diameter decreased from 13.1 to 8.7 μm. To have a clearer comparison with similar diameter about 10μm, the tensile strength of ABF-CF and NDP-CF were 860 MPa and 640 MPa, respectively, and the modulus were 41 GPa and 33 GPa, respectively. We have added more conclusions and discussion in the revision (Line 242-250)

(Comment 11)

Captions do not provide sufficient details of the images in the figures.

(Answer to comment 11)

We have revised some captions of tables and figures to provide sufficient details (lines 144, 146-147, 213, 234).

Reviewer 2 Report

This paper showed that a comparison of ethylene tar-derived isotropic pitches prepared by air blowing and nitrogen distillation method and their carbon fibers. This article is very interesting but there are some observations to make the article more complete.

Line 52, 53: Please describe more detail why the evolution and influence of oxygen was ignored. If influence of oxygen is considered to process of CF, describe to give any effects in CF industry.

Line 65: For oxygen effect, the authors conducted several tests. If there are test standards for each experiment, please add a reference for standard test.

Author Response

Response to reviewer 2

Dear Editor

• Manuscript No.: Materials-413012

• Manuscript Title: A comparison of ethylene tar-derived isotropic pitches prepared by air blowing and nitrogen distillation method and their carbon fibers

• Manuscript Authors: Kui Shi, Jianxiao Yang*, Chong Ye, Hongbo Liu and Xuanke Li*

We would like to express our sincere thanks to the reviewers for their valuable comments and constructive suggestions. We have tried our best to revise our manuscript according to their suggestions. We uploaded the revised manuscript which we would like to be accepted to publish in Materials. Revised parts in the manuscript were highlighted in YELLOW.

Response to Reviewer 2:

We would like to express our sincere thanks to this reviewer for her/his positive evaluation, valuable comments and constructive suggestions.

(Comment 1)

Line 52, 53: Please describe more detail why the evolution and influence of oxygen was ignored. If influence of oxygen is considered to process of CF, describe to give any effects in CF industry.

(Answer to comment 1)

There were some researches on air blowing of pitches mainly focused on the influence of the different raw materials, for instance, coal tar pitch, petroleum pitch and anthracene oil or diverse conditions of air blowing on the final various properties of air blown pitches. However, they only investigated the pitches and rarely referred to the preparation of carbon fibers. As a matter of fact, there is also oxygen introduced during stabilization process to make the pitch fibers infusible and maintain the fiber shape during carbonization process of preparing carbon fibers besides air blowing process, and the oxygen introduced in these two stages may have different behavior during subsequent carbonization process. Therefore, the objective of our work is to make it clear that whether the oxygen introduced by air blowing during preparation of pitches can promote the stabilization of pitch fibers and improve the properties of carbon fibers (lines 51-58).

(Comment 2)

Line 65: For oxygen effect, the authors conducted several tests. If there are test standards for each experiment, please add a reference for standard test.

(Answer to comment 2)

Thanks for your constructive suggestions. We have conducted several tests on pitches and fibers to obtain their properties. Some characterization is according to standard, for example, solubility parameter (GB/T 26930.5-2011), elemental analysis (SN/T 4764-2017) and mechanical properties (ASTM D4018-2011), these standards are supplemented in the section 2.4 (lines 94-95, 97, 121). Some measurements such as FT-IR, 13C-NMR and XRD are generic methods in the field of pitch and carbon fiber and no definite standard. There characterization and analysis refer to other paper in Ref [23, 25, 26].

Reviewer 3 Report

I would recommend getting it reviewed by a English language write once more to fine tune language and flow of work. Please  provide higher magnification of carbonized carbon fiber to show graphetization How does this cross section compare to commercial carbon fiber. Was any composite prepared to see inter-facial adhesion between matrix and fibers? If not please provide data Otherwise great work  keep it up
